# Speckle Plethysmograph-Based Blood Pressure Assessment

Floranne T. Ellington [1], Anh Nguyen [2], Mao-Hsiang Huang [1], Tai Le [3], Bernard Choi [3,4] and Hung Cao [1,2,3,*]

1    Department of Electrical Engineering and Computer Science, University of California Irvine,
     Irvine, CA 92697, USA; fellingt@uci.edu (F.T.E.); maohsiah@uci.edu (M.-H.H.)
2    Department of Computer Science, University of California Irvine, Irvine, CA 92697, USA; anhtn18@uci.edu
3    Department of Biomedical Engineering, University of California Irvine, Irvine, CA 92697, USA;
     tail3@uci.edu (T.L.); choib@uci.edu (B.C.)
4    UCI Beckman Laser Institute & Medical Clinic, University of California Irvine, Irvine, CA 92697, USA
*    Correspondence: hungcao@uci.edu

**Abstract:** Continuous non-invasive blood pressure (CNBP) monitoring is of the utmost importance in detecting and managing hypertension, a leading cause of death in the United States. Extensive research has delved into pioneering methods for predicting systolic and diastolic blood pressure values by leveraging pulse arrival time (PAT), the time difference between the proximal and distal signal peaks. The most widely employed pairing involves electrocardiography (ECG) and photo-plethysmography (PPG). Possessing similar characteristics in terms of measuring blood flow changes, a recently investigated optical signal known as speckleplethysmography (SPG) showed its stability and high signal-to-noise ratio compared with PPG. Thus, SPG is a potential surrogate to pair with ECG for CNBP estimation. The present study aims to unlock the untapped potential of SPG as a signal for non-invasive blood pressure monitoring based on PAT. To ascertain SPG's capabilities, eight subjects were enrolled in multiple recording sessions. A third-party device was employed for ECG and PPG measurements, while a commercial device served as the reference for arterial blood pressure (ABP). SPG measurements were obtained using a prototype smartphone-based system. Following the completion of three scenarios—sitting, walking, and running—the subjects' signals and ABP were recorded to investigate the predictive capacity of systolic blood pressure. The collected data were processed and prepared for machine learning models, including support vector regression and decision tree regression. The models' effectiveness was evaluated using root-mean-square error and mean absolute percentage error. In most instances, predictions utilizing $PAT_{SPG}$ exhibited comparable or superior performance to $PAT_{PPG}$ (i.e., SPG Rest $\pm$ 12.4 mmHg vs. PPG Rest $\pm$ 13.7 mmHg for RSME, and SPG 8% vs. PPG 9% for MAPE). Furthermore, incorporating an additional feature, namely the previous SBP value, resulted in reduced prediction errors for both signals in multiple model configurations (i.e., SPG Rest $\pm$ 12.4 mmHg to $\pm$3.7 mmHg for RSME, and SPG Rest 8% to 3% for MAPE). These preliminary tests of SPG underscore the remarkable potential of this novel signal in PAT-based blood pressure predictions. Subsequent studies involving a larger cohort of test subjects and advancements in the SPG acquisition system hold promise for further improving the effectiveness of this newly explored signal in blood pressure monitoring.

**Keywords:** blood pressure; continuous non-invasive methods; pulse arrival time; speckleplethys-mography

## 1. Introduction

Blood pressure (BP) is a critical parameter for assessing patient well-being. BP monitoring is essential to reduce mortality and morbidity. BP consists of systolic (SBP) and diastolic (DBP) values, which reflect arterial pressure during heartbeats and rest periods. Typically, SBP and DBP should be below 120 mmHg and 80 mmHg, respectively. When SBP and DBP are consistently 130 mmHg and 80 mmHg or higher, this indicates high BP (HBP) or hypertension, which increases the risk of heart attacks and stroke—two leading causes

of mortality in the U.S. Thus, continuous BP measurement is essential in daily settings for at-risk individuals.

BP is a dynamic physiological parameter that changes over time due to factors such as age, activity, and mental stress [1,2]. Non-continuous BP measurements cannot fully reveal these dynamic characteristics of BP on individuals; thus, continuous BP monitoring can become much more informative if it is widely available and easy to obtain. Furthermore, continuous measuring sensors can be feasibly designed to be cuffless and wearable. The Penaz method, also known as a volume clamp, and tonometry [3,4] are popular continuous, non-invasive BP methods. The Penaz method is cuff-based and optically measures the arterial volume in a limb, such as a finger or a toe, by applying pressure via an occluding cuff. The accuracy of volume clamp methods is known to be sensitive to the automatic recalibration process, resulting in overestimated SBP frequently [3]. Additionally, continual use of volume clamps increases the risk of venous congestion in the measuring site, and repeated and long-term wear to the same region become very uncomfortable and even painful for the subject, making this method not viable for long-term wear [3,5]. Tonometry measures arterial pressure by applying force over a superficial artery to distort the vessel. This can be performed through a wristband or with a hand-held instrument and records pulsatility from applied force flattening the chosen superficial artery, which overcomes issues with blood vessel occlusion [4]. However, tonometry methods can become problematic as they are sensitive to imprecise placements of the device and easily result in inaccurate readings, especially with patient movement. Considering this, establishing a calibrated baseline of tonometry readings remains challenging and currently not a viable path for CNBP.

Currently, the best and most common home-based BP monitoring technology has been using commercial BP cuffs [6]. However, as previously stated, CNBP cannot be realistically achieved with this method. Recently, alternative methods have been explored for continuous indirect BP monitoring using signals such as electrocardiography (ECG) and photoplethysmography (PPG) [1–5,7,8]. ECG is a widely utilized biosignal within the medical domain, particularly in clinical contexts, for diagnosing cardiovascular conditions and monitoring vital signs [9]. PPG constitutes an optical measurement frequently employed in pulse oximetry within clinical environments to assess oxygen saturation levels. This optical methodology gauges blood flow in a region of interest (ROI) by measuring optical absorption or reflection along the optical path. Essentially, PPG detects changes in blood volume using a photoelectric technique [10]. PPG sensors are categorized into two main measurement configurations: transmission mode and reflection mode. In the transmission mode, the setup includes a light-emitting diode (LED) positioned on one side of the tissue of the ROI, with a photodetector (PD) on the opposing side. Here, the light emitted traverses through the tissue, modulated by the underlying vasculature, and the modulated optical energy is then detected on the other side. Conversely, in the reflection mode, both components (LED and PD) are situated on the same side of the tissue, typically on the same plane. The optical signal penetrates the tissue, and the PD receives the reflected light with fluctuations due to tissue absorption. While the transmission mode is primarily limited to the earlobe, fingertip, and toe, the reflection mode extends to additional locations as long as the ROI is a flat area (e.g., the forehead, forearm, supraorbital artery, under the legs, and the wrist) [11].

Pulse arrival time (PAT), representing the time it takes for the heartbeat peak to reach the peripheral ROI, enables non-invasive BP estimation through linear and non-linear equations and supervised machine learning regression models [12–14]. PPG and ECG are a greatly explored pair using this technique; however, recently, a new biosignal similar to PPG has surfaced in research around non-invasive measurements and has been under-investigated.

Speckleplethysmography (SPG) has emerged as a promising alternative to PPG for measuring heart rate variability, microvascular flow, and resistance [15–19]. SPG relies on laser speckle contrast imaging (LSCI) to monitor blood flow changes [20], also known as

Affixed Transmission Speckle Analysis (ATSA) [21]. SPG devices employ a laser source emitting rays through tissue thickness into a CMOS camera on the opposite side or on the same side similar to PPG's transmission and reflectance modes. The camera captures a raw video with a minimum of 30 frames per second (fps) and applies image processing techniques to identify speckles from the laser within the red channel of each frame. The quantity of speckles detected correlates with the blood flow at that time point, and their temporal changes constitute the SPG waveform [22–24]. This method is a recently realized waveform to measure heart rate variability [19]. SPG offers advantages like higher signal-to-noise ratio (SNR) and consistent performance across temperatures [25] as well as potential for wearables adaptation due to its promising results using reflectance mode compared to PPG [26]. Our study explores the potential of SPG for PAT-based BP monitoring. We used machine learning models to compare SPG and PPG techniques and proved that SPG signals from off-the-shelf smartphones can non-invasively monitor accurate blood pressure in real-world scenarios, supported by empirical evidence that shows solid correlations between PAT measurements obtained from SPG signals and reference blood pressure measurements obtained using standard clinical methods. Our work showcases the value of SPG-based blood pressure estimation systems in the realm of telemedicine, remote patient monitoring, and personalized healthcare, with the practical considerations and challenges that come with adopting and deploying SPG-based blood pressure estimation systems in clinical and home settings.

## 2. Materials and Methods

An in-house system was developed to effectively collect SPG, ECG, PPG, and arterial BP (ABP) concurrently (Figure 1A). To capture SPG, a smartphone camera (Samsung Galaxy A10e rear camera) and a 635 nm red laser diode were used (Figure 1C). ECG and PPG were recorded using a third-party device (AFE4950EVM from Texas Instruments). The ABP readings were acquired with an Omron inflatable cuff for post-experimental accuracy metrics.

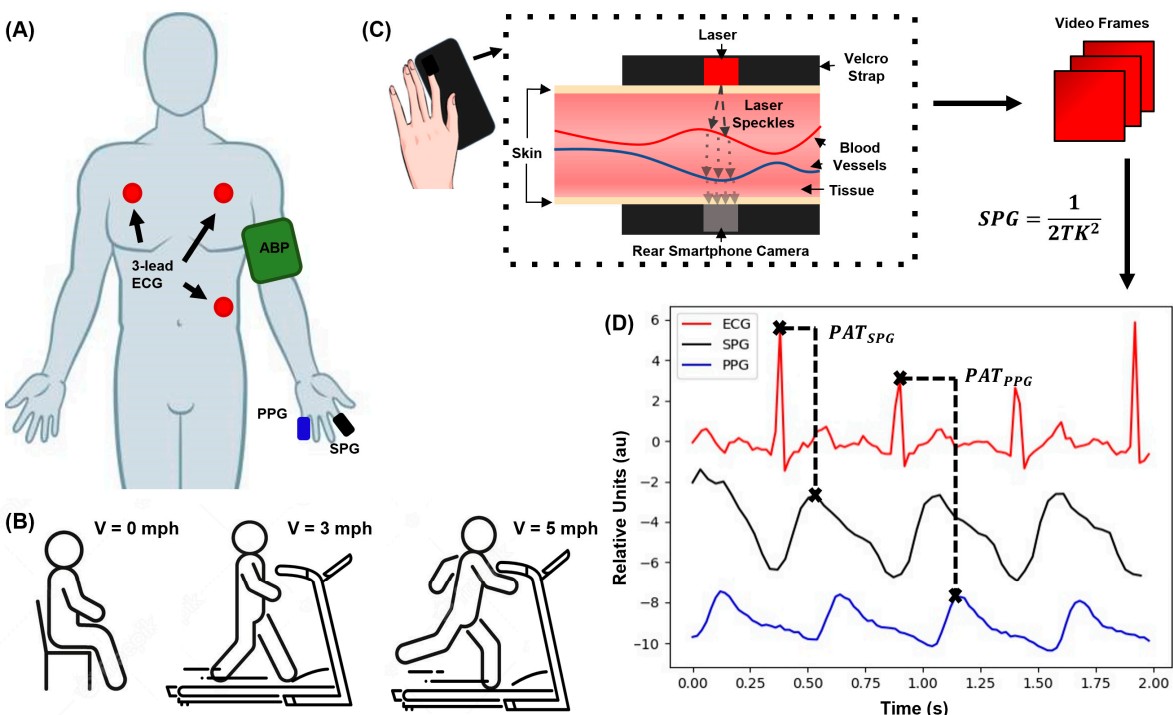

**Figure 1.** Experimental setup, SPG acquisition diagram, and the acquired signals. (**A**) Acquisition illustration for the 4 sensors, ABP, ECG, PPG, and SPG, and the locations where they are recorded,

respectively. (**B**) Activities each subject completes per session including (from left to right): (i) sitting at rest, (ii) walking on a treadmill at 3 mph (LE), and (iii) running on the treadmill at 5 mph for 2.5 min (HE). After each activity, the subject's signals are immediately recorded. (**C**) The phone-based transmission-mode SPG system and signal processing method diagram. (**D**) The SPG signal in the time domain with the corresponding ECG and PPG signals. The panel shows how $PAT_{PPG}$ and $PAT_{SPG}$ are obtained.

### 2.1. Experimental Procedure

Each session involved three stages: Resting for 2.5 min (Rest); walking at 3 mph on a treadmill for 2.5 min (Light Exercise—LE); and running at 5 mph for 2.5 min (Heavy Exercise—HE) (Figure 1B). The treadmill was used to standardize the impact of the activities across all subjects. After completing each stage, the subject's ABP was measured using the Omron device, positioned on their left upper arm. Simultaneously, the subject had their fifth and second digits attached to the PPG and SPG sensors, respectively, using their left hand (Figure 1A). PPG was recorded using reflectance mode, while SPG was recorded in transmission mode due to the nature of their respective measurement systems. ECG measurements were obtained through a 3-lead chest configuration employing snap electrodes affixed to the subject's chest. The ECG signal was sampled at a frequency of 500 Hz, while the PPG signal was sampled at 50 Hz, which were the frequency sampling rates set by the TI board. These signals were then saved to a .csv file on the computer connected to the TI board. The SPG signal was sampled at a frame rate of 30 fps, which was the maximum achievable with the A10e Samsung smartphone. These signals were monitored for a duration of 60 s while the subject maintained a relaxed posture and were synchronized using timestamps from both devices.

### 2.2. Data Processing and Models

A Python script was developed to process the video and BP data [26]. The .csv file containing the ECG and PPG data is read and reworked into data frames with their timestamps. The SPG was derived by extracting the red intensity of each frame from the videos, and is calculated as [19]:

$$SPG = \frac{1}{2TK^2} \tag{1}$$

where $T$ represents the exposure time of the image, and $K$ denotes the average speckle contrast squared. These values plus their timestamp are saved to another data frame. If needed, all the signals have their wandering baseline removed using an open-source algorithm, and a Savitzky–Golay filter is applied to smoothen the signal, similar to [27]. The filtered ECG and PPG signals were downsampled to match the rate of the SPG recordings. They were exhibited alongside the SPG signal to facilitate subsequent PAT computations (Figure 1D). The signals were plotted in 10 s windows and analyzed by hand.

In this study, we employed two supervised machine learning regression models: support vector (SV) regression and decision tree (DT) regression. SV regression seeks the best-fitting line or hyperplane by maximizing the margin between data points and the regression line, while DT regression partitions the feature space to predict the target variable based on average values within each region. These models were selected based on their reported improved accuracy and efficacy [13,14]. The first feature employed in the models is the PAT obtained from the respective signal pairs ($PAT_{signal}$), while the second feature is the previous SBP value ($SBP_{n-1}$). Each model was assessed using two distinct feature configurations: one exclusively utilizing $PAT_{signal}$ and the other comprising $SBP_{n-1}$ and $PAT_{signal}$. Due to the two PAT pairs, two different ML models were utilized, as well as the different features incorporated. Between the three exercise datasets, 24 total models were trained, tested, and compared for accuracy between the two signal methods.

## 3. Results

In total, nine data points were recorded from each of the eight subjects, corresponding to the three performed activities over three sessions. For consistency, the three sessions took place either in the morning or evening, with a time difference of less than 48 h between each subject's first and last sessions. The sessions consisted of the same format and order of activities: Rest, LE, and HE. For each exercise dataset, 24 one-minute sessions were recorded and used for their respective models. Different testing times were chosen to observe variations in BP throughout the day. The subjects' ages ranged from 21 to 30 years of age, with various levels of fitness. Five subjects lived a sedentary lifestyle, two infrequently visited the gym, and one regularly worked out five or more times a week. Two of the eight subjects were female, while the rest were male, and only one subject had prior hypertension-related issues. The data recorded from this individual did not significantly deviate from the other data points and was thus included in the final datasets. Additionally, the individuals were asked about their amount and quality of sleep as well as caffeine intake before each session. Seven out of eight subjects reported having good sleep prior to the test, and six out of eight subjects consumed caffeine earlier that day before the experiment. Between the three sessions per subject, no significant differences in BP values and signals were noted. Fitness level did result in different average heartrates and BP values but not enough to greatly influence the end results. Additionally, sleep quality and caffeine consumption did not result in consistent outlier data.

From these sessions, PAT was derived from the ECG, PPG, and SPG signal peaks. A distinct segment of the signals (Figure 1D) was selected to determine PAT by calculating the peak-to-peak differences between ECG and PPG, as well as the peak-to-peak between ECG and SPG. Some measurements resulted in unclear PPG or SPG signal peaks and were consequently excluded from further consideration. The data points were then categorized into three separate datasets based on the activity performed by the subject before measurement: Rest, LE, and HE.

Prior to training SV and DT models, the datasets for SBP, SPG-based PAT ($PAT_{SPG}$), and PPG-based PAT ($PAT_{PPG}$) were scrutinized for outliers using box plots. The total numbers of data points per dataset after removing outliers was 18, 17, and 19 for Rest, LE, and HE, respectively. Given the relatively small size of the dataset and the need to ensure robust assessment of the models' performance, the testing sample for each dataset was nine of the data points from one random subject. The models were trained using all but nine data points from the random subject, six of the nine data points were utilized as a validation dataset to choose the proper models, and the remaining subjects' data were reserved for testing the model's performance. This was used to essentially mimic the leave-one-out cross-validation scheme small datasets use to combat and mitigate overfitting. Root-mean-square errors (RMSEs) and mean absolute percentage errors (MAPEs) were calculated to evaluate the predictions of the regression models (Tables 1 and 2). Bland–Altman plots were generated, revealing a strong agreement between PPG and SPG in PAT trends (Figure 2). The *p* values of each activity dataset were calculated and resulted in $p < 0.0001$ for all three datasets, further demonstrating strong evidence for using SPG in place of PPG for blood pressure estimations. Additionally, Figure 3 demonstrates the predicted results (the dots) compared to the standard deviation (the range) on the mean (the end of the bar plots).

**Table 1.** The root-square-mean error (RSME) comparison of SPG and PPG using PAT and previous BP measurements for features in supervised machine learning models. Values are bolded when they demonstrate SPG performing better or the same as PPG.

| RSME (± mmHg) | SV (1 Feat.) | SV (2 Feat.) | DT (1 Feat.) | DT (2 Feat.) |
|---|---|---|---|---|
| **SPG Rest** | **12.4** | **3.7** | **3.9** | 2.8 |
| **PPG Rest** | 13.7 | **3.7** | 22.5 | 1.9 |

**Table 1.** *Cont.*

| RSME (± mmHg) | SV (1 Feat.) | SV (2 Feat.) | DT (1 Feat.) | DT (2 Feat.) |
|---|---|---|---|---|
| SPG LE | 16.1 | **15.9** | 20.1 | 2.2 |
| PPG LE | 15.3 | **15.9** | 14.1 | 3.7 |
| SPG HE | **20** | **19.3** | 22.7 | 25.9 |
| PPG HE | 21 | **19.3** | 11.9 | 17.2 |

**Table 2.** The mean absolute percent error (MAPE) comparison of SPG and PPG using PAT and previous BP measurements for features in supervised machine learning models. Values are bolded when they demonstrate SPG performing better or the same as PPG.

| MAPE (%) | SV (1 Feat.) | SV (2 Feat.) | DT (1 Feat.) | DT (2 Feat.) |
|---|---|---|---|---|
| SPG Rest | **8** | **3** | **3** | 2 |
| PPG Rest | 9 | **3** | 15 | 1 |
| SPG LE | 11 | **10** | 14 | **2** |
| PPG LE | 10 | **10** | 9 | 3 |
| SPG HE | **10** | **11** | 11 | 14 |
| PPG HE | 11 | **11** | 7 | 9 |

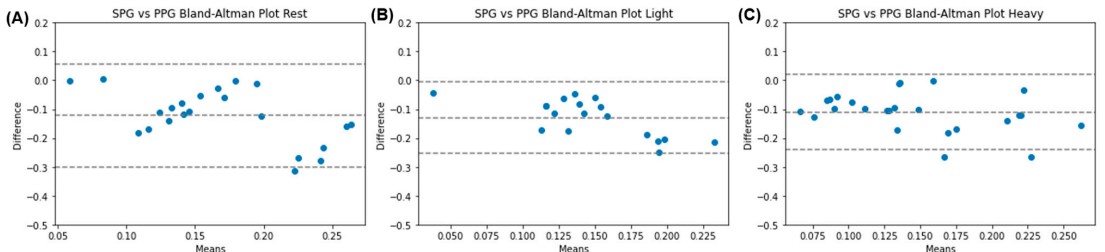

**Figure 2.** Bland–Altman plots showing the agreement between $PAT_{SPG}$ and $PAT_{PPG}$ for (**A**) Rest data; (**B**) Light Exercise (LE) data; and (**C**) Heavy Exercise (HE) data.

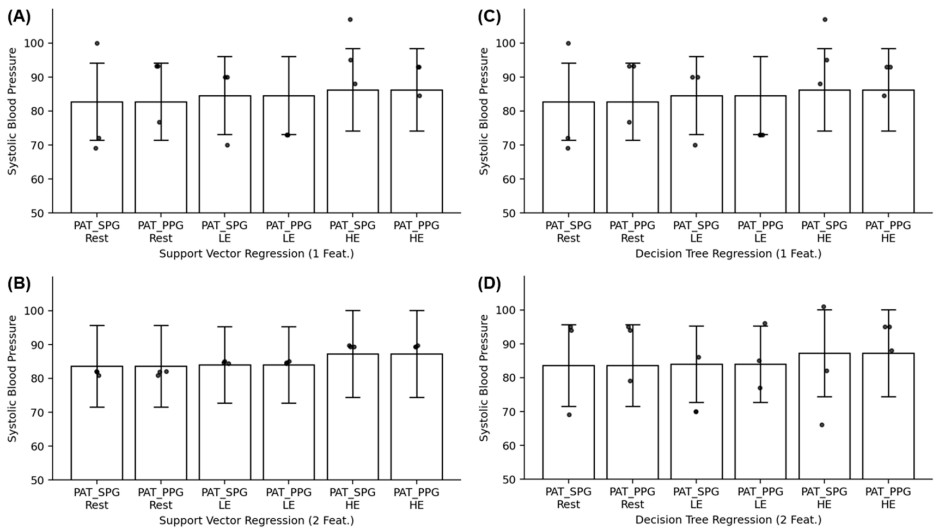

**Figure 3.** Error bar plots of $PAT_{SPG}$ and $PAT_{PPG}$ for each activity dataset for (**A**) SV regression using 1 feature, (**B**) SV regression using 2 features, (**C**) DT regression using 1 feature, and (**D**) DT regression using 2 features.

## 4. Discussion

The study demonstrates SPG's potential as a substitute for PPG in PAT-based BP estimation. SPG generally yields comparable or smaller errors than PPG in predicting SBP, as seen by the bolded values in Tables 1 and 2. Even in the other results where PPG has a lower error, SPG performs decently in comparison. For the SV models, SPG predicts within 1.5 mmHg for RSME and 1% for MAPE compared to the PPG results. From the DT models, the error is higher but lessens when the second feature is included in training. Incorporating the previous SBP value reduces the RMSE, particularly in the Rest dataset with the SV model and the LE dataset with the DT model (Table 1), indicating potential accuracy improvement. This feature is essentially acting as a form of calibration for the model and will be investigated further in a future study. For the SV models, this second feature unexpectedly resulted in the same RSME and MAPE values as PPG, indicating how the previous blood pressure value greatly influences the model's predictions.

The SV model's robustness against overfitting and reliance on the previous SBP value explain its significant improvement on the Rest dataset in both tables [28]. The DT model's reduced error using two features for the LE dataset may be due to overfitting tendencies exacerbated by the smaller LE dataset due to more outliers [29]. Additionally, this overfitting may have contributed to the wide range of results for both SPG and PPG over all three activity datasets. Model selection and feature inclusion highly influence BP estimation accuracy for SPG and PPG.

RMSE and MAPE values increase with higher physical activity levels, elevated heart rate, and blood pressure due to the increase in motion artifacts that negatively affect BP estimation accuracy. SV models show a uniformed error increase within acceptable values, while DT models exhibit a more arbitrary and pronounced increase, highlighting different evaluation approaches and DT's susceptibility to overfitting with small datasets [29]. Figure 3 further demonstrates DT's overfitting, as some of the results for both feature configurations deviate more than the standard deviation of the dataset. The unexpected error spike in the PPG Rest dataset (Tables 1 and 2) when using only one feature emphasizes the importance of incorporating a second feature, which reduced error significantly, illustrated by Figure 3B in all dataset results. Feature selection is crucial for accurate BP estimation using SPG and PPG, and the diverging error trends between the models using small datasets highlight the need for careful model selection and evaluation, considering their strengths and weaknesses.

The Bland–Altman plots reveal a negative mean difference between the $PAT_{SPG}$ and $PAT_{PPG}$ datasets, which is due to SPG peaking before PPG despite being recorded from the same region of interest. The limits of agreement tighten from Rest to HE, which might indicate overfitting being the reason that LE and HE had increasingly higher errors compared to the Rest based models. Over the three datasets, the Bland–Altman plots have a mean difference within 0.12 of zero, indicating that $PAT_{SPG}$ can potentially be a reliable and consistent alternative to $PAT_{PPG}$ for future blood pressure estimation. Furthermore, each activity dataset resulted in $p < 0.0001$, demonstrating the results were not due to random chance and are statistically significant findings.

Limitations of the SPG system used include the absence of real-time SPG signal visualization, leading to data collection issues and unusable data points for the datasets. Signal synchronization relied on timestamps and knowledge of SPG peak timing relative to PPG and ECG. Stable hand positioning was also required for signal quality. These limitations can be addressed by developing a wearable system enabling simultaneous measurement of all signals, enhancing real-time visualization, signal acquisition quality, and synchronization. Due to the current technological developments of smart watches, cloud computing, and smaller chips, a custom wearable can be developed and validated for future experiments. The relatively small experiment size suggests the need for larger studies with at least 50 subjects to examine the impact of daily activities on BP, avoid overfitting issues, and obtain more reliable and generalizable results. Furthermore, the lack

of difference between morning and evening sessions needs to be investigated more with a larger group to determine its role on BP and the biosignals.

Using PAT versus pulse transit time (PTT) is also a topic to investigate with SPG-based BP estimation. PTT is the time that an arterial pressure wave takes to propagate along the walls of a given segment of the arterial tree. PAT is defined as the aggregate of PTT and the pre-ejection period (PEP) delay, expressed as:

$$PAT = PTT + PEP \qquad (2)$$

Here, PEP represents the duration required for converting the electrical signal into a mechanical pumping force, leading to isovolumetric contraction and subsequent opening of the aortic valve. PEP can be quantified by determining the delay between the R-wave of the ECG signal [3,4]. Given the intricacy involved in acquiring PEP when multiple physiological signals are necessary, the utilization of PAT to estimate BP has gained traction. However, the precision of PAT-based BP estimation remains a subject of debate [30]. Due to this, comparing the PAT and PTT derived from the signal pairs may reveal how inaccurate PAT is for SPG-based BP estimation and whether the ease of calculating this value compared to PTT is worth the potential loss of accuracy.

Another point that needs to be investigated further is whether to focus on a calibrated system or develop a calibration-free system. This study had a form of calibration by using the subject's previous SBP value in training and testing the models. This helped improve the accuracy for all SVM results and all but the HE results in the DT models. For real-time deployment of the system, a small calibration period could be used to benefit from higher accuracy like this setup, but calibration-free systems may prove to be more user friendly and have less computation cost and time.

## 5. Conclusions

This study highlights the potential use of SPG as a viable alternative to PPG in PAT-based BP estimation. The agreement observed in the Bland–Altman plots (Figure 2) further strengthens the case for the reliability and accuracy of SPG measurements. The inclusion of additional features and careful model selection have demonstrated their influence on the accuracy of BP estimation using SPG and PPG. However, it is crucial to address the limitations of the SPG system, such as the absence of real-time signal visualization and signal synchronization challenges. Developing a wearable system capable of simultaneous signal acquisition would overcome these limitations and enable more reliable and convenient BP measurements during various activities.

Future research should focus on expanding the sample size to ensure the robustness and generalizability of the results. Incorporating other features, such as the SPG intensity ratio feature, and adopting dynamic adaptive regression models tailored to different activity levels are potential avenues for higher accuracy of BP estimation using both PPG- and SPG-based PAT, capturing the intricacies of BP regulation and the cardiovascular system. Further investigations should also explore additional features and algorithmic improvements to advance the field of non-invasive BP estimation and its clinical applications.

**Author Contributions:** Conceptualization, F.T.E., B.C. and H.C.; methodology, F.T.E. and T.L.; software, F.T.E. and A.N.; validation, F.T.E. and A.N.; formal analysis, F.T.E.; investigation, F.T.E.; resources, H.C.; data curation, F.T.E. and A.N.; writing—original draft preparation, F.T.E.; writing—review and editing, A.N., F.T.E., T.L., B.C., M.-H.H. and H.C.; visualization, F.T.E. and M.-H.H.; supervision, H.C.; project administration, H.C.; funding acquisition, H.C. All authors have read and agreed to the published version of the manuscript.

**Funding:** This research was sponsored under a seed grant from the Office of Research at UC Irvine.

**Institutional Review Board Statement:** Ethical review and approval were waived for this study due to subjects being from the same research group as the first author.

**Informed Consent Statement:** Informed consent was obtained from all subjects involved in the study. Written informed consent has been obtained from the patient(s) to publish this paper.

**Data Availability Statement:** The data presented in this study are available on request from the corresponding author.

**Acknowledgments:** One of the authors wants to acknowledge the GAANN fellowship support from the Dept. of Education under award P200A180052 for support of this study.

**Conflicts of Interest:** The authors declare no conflicts of interest. The funders had no role in the design of the study; in the collection, analyses, or interpretation of data; in the writing of the manuscript; or in the decision to publish the results.

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
