# Peer review of "Speckle Plethysmograph-Based Blood Pressure Assessment"

_technologies, doi:10.3390/technologies12050070_

Round 1
Reviewer 1 Report
Comments and Suggestions for Authors
This paper evaluates several machine learning algorithms for blood pressure supervision. The paper and its application are interesting, but the contribution is low. The number of subjects included in the analyses is low and the number of analyses is also low.
Comments to improve the paper:
· I’d suggest including a list of contributions at the end of the introduction.
· I’d suggest including a SOTA section.
· Regarding the experimental procedure, I have seen that you are training and testing with examples from the same subject. It means that the system is a subject dependent system. I think this aspect must be commented in detail in the paper as a limitation, or consider the possibility to do several subject independent experiments.
· I have seen that you did not consider a validation subset. How did you finetune the hyperparameters of the ML algorithms?
· I miss a stronger comparison with previous works. I’d suggest including a table with figures from previous works.
· I’d suggest including statistical analyses of the results.
Reviewer 2 Report
Comments and Suggestions for Authors
I have the following comments about this paper:
1. The authors may need to quantify the added value of including the previous SBP value in different model configurations with a thorough analysis or statistical comparison in the results section. How did the addition of the systolic blood pressure (SBP) value specifically contribute to the observed improvements in prediction accuracy for both SPG and PPG?
2. The author should explain the difficulties posed by inaccurate tonometry device placement and patient movement, and explore other approaches are being investigated or developed at this time to create a trustworthy baseline for continuous non-invasive blood pressure (CNBP) monitoring.
3. The author may cover the particular features or attributes of SPG. What make it a potentially useful supplement to the current array of non-invasive blood pressure measurement instruments? Furthermore, in comparison to recognized techniques like PPG and ECG, what are the preliminary discoveries or difficulties found in the initial research that call for additional research into its accuracy and efficacy?
4. When derived from signal pairs, how significant are the differences in accuracy between PAT-based and PTT-based blood pressure estimations, and in what specific conditions does one method regularly perform better than the other?
5. For the discussion part, the author may need to discuss the technological developments or breakthroughs are required to create a wearable system that can acquire signals for SPG and PPG simultaneously and debate the real-time signal display and efficient synchronization. For, example, how could a system of that kind manage the computational load while maintaining precision and dependability in a variety of physical tasks and environmental circumstances?
Reviewer 3 Report
Comments and Suggestions for Authors
Review of the manuscript technologies-2921175: Speckle Plethysmograph-based Blood Pressure Assessment
This paper evaluated speckle plethysmography as an alternative to photo-plethysmography in blood pressure estimation. Experimental data were collected to compare the two methods. The design of the experiment is good, and the introduction is comprehensive. Also, the promising results suggest that this method deserves further investigation.
I recommend better introducing SV and DT methods to support the discussion but also the rationale for selecting specifically these TWO ML methods. Also, the analysis of the data and the discussion could be improved according to the following list of comments:
Line 152: The two selected ML methods are introduced in only one sentence. The paper would benefit from briefly describing regressions by support vector and decision tree.
Line 209 and 218: please add some references about the SV model’s robustness and DT’s susceptibility to overfitting.
Line 219: “Unexpected error spike in the PPG rest dataset”: this statement could come with graphs or illustrations retrieved from the dataset—the same comment for overfitting issues.
Tables 1 and 2: I recommend discussing further these two tables. As an example, DT shows increased RSME and MAPE for SPG HE and PPG HE. Also, SV results through 1 or 2 features often yield similar results. Did the authors expect such results?
Statistical analysis: Bland-Altman plots are useful; however, standard statistical tests may provide better confidence in the significance of the claimed results.
Round 2
Reviewer 1 Report
Comments and Suggestions for Authors
The authors have addressed my comments and the paper can be accepted.
Reviewer 3 Report
Comments and Suggestions for Authors
I would like to thank the authors for conscientiously taking into account all the comments made during the revision of the manuscript. I recommend the publication of this interesting paper in Technologies.